# Bioinformatics Tools and Approaches for Virus Discovery in Genomic Data: A Systematic Review

**DOI:** 10.3390/v17121538

**Published:** 2025-11-24

**Authors:** Julia Galeeva, Polina Kuzmichenko, Alexander Manolov, Alexander Lukashev, Elena Ilina

**Affiliations:** 1Research Institute for Systems Biology and Medicine (RISBM), Department of Mathematical Biology and Bioinformatics, Moscow 117246, Russia; kuzmichenko.p.a@gmail.com (P.K.); paraslonic@gmail.com (A.M.); alexander_lukashev@hotmail.com (A.L.); ilinaen@gmail.com (E.I.); 2Martsinovsky Institute of Medical Parasitology, Tropical and Vector Borne Diseases, Sechenov University, Moscow 119435, Russia

**Keywords:** viruses, taxonomic annotation, HMM, machine learning, metagenomics, bioinformatics, deep learning, viral classification

## Abstract

The exponential growth of viral metagenomic data has created an urgent need for accurate and scalable tools for virus discovery, yet the extreme diversity, rapid evolution, and limited reference databases for viruses pose unique computational challenges that traditional sequence comparison methods struggle to address. This systematic review, conducted in accordance with PRISMA 2020, examines current trends and methodological advances in virus discovery tools from 1990 to 2025. As virus discovery is a broad and multi-dimensional topic, this review focuses on the first-line tools used to analyze the results of high-throughput sequencing. The review was conducted using the PubMed database with a snowballing approach, with over 54 key studies selected for the analysis. These studies encompass the following approaches: alignment-based methods, rapid similarity estimation techniques, profile hidden Markov model methods, combination pipelines, k-mer-based approaches, and machine learning-based methods. The transition from alignment-based to machine learning methods has dramatically improved the detection of divergent viruses, yet challenges remain in interpreting model decisions and handling incomplete viral genomes. This review summarizes current knowledge and potential future directions for the development of virus detection capabilities.

## 1. Introduction

Viruses are the most abundant biological entities on Earth, with an estimated 10^31^ particles globally, colonizing every environment where potential hosts exist [1]. Viral abundance varies by habitat: marine samples contain 8.5 × 10^5^ to 2.2 × 10^7^ virus-like particles (VLPs) per mL, soil environments show densities of 10^7^–10^9^ VLPs/g and account for 90–95% of global viral biomass [2,3]. The variety of viral hosts is immense, spanning all domains of life: bacteria, archaea, protists, fungi, plants, and animals, including humans.

Viruses exhibit exceptional diversity in their genetic material organization. Viral genome sizes range from 1.8 kb to 2.5 Mb, reflecting a broad spectrum of genetic information encoding strategies [4]. Current classification of viruses is based on host cell type: viruses of eukaryotes, viruses of archaea (archaeal viruses), and viruses of bacteria (bacteriophages); nucleic acid type and organization (DNA-containing deoxyviruses and RNA-containing riboviruses, single- and double-stranded, linear and circular); and structural organization (enveloped and non-enveloped viruses). Viruses drive host evolution through horizontal gene transfer and dramatically alter cellular metabolism—cyanophages reprogram photosynthesis in *Synechococcus*—while auxiliary metabolic genes enhance host fitness under stress [5]. Many phages persist as integrated prophages, conferring capabilities from toxin production to antibiotic resistance [6].

Despite their abundance and importance, virus classification faces fundamental challenges. Unlike cellular organisms with universal ribosomal RNA markers, viruses lack conserved genes across all lineages [6,7]. Combined with extreme mutation rates (10^−6^ to 10^−4^ substitutions per nucleotide per cell in RNA viruses), extensive horizontal gene transfer [8], and polyphyletic origins [9], viral genomes represent evolutionary mosaics that challenge traditional taxonomy. Nevertheless, viral taxonomy continues to expand rapidly: as of 2025, the International Committee on Taxonomy of Viruses (ICTV) has established 16,215 virus species organized into seven realms, 11 kingdoms, 22 phyla, and numerous lower taxonomic ranks, with recent additions including the new phylum *Ambiviricota* for viruses combining features of both typical RNA viruses and viroids [10].

The study of viral diversity remains severely limited by methodological constraints. Traditional virology depends on cultivating viruses from cultured hosts, yet most bacteria, and thus their phages, cannot be grown in laboratory settings. While next-generation sequencing enables culture-independent discovery, known viruses remain underrepresented in metagenomic datasets, typically comprising less than 5% of sequencing reads due to their small genomes. Most significantly, the “viral dark matter” persists—40–90% of viral sequences in metagenomic studies show no homology to known viruses [11,12], reaching 95% in environmental samples. The rapid expansion of metagenomic sequencing has created an urgent need for accurate virus discovery tools, yet the extreme diversity and rapid evolution of viruses pose unique computational challenges.

The identification of viruses in metagenomic datasets has undergone a remarkable transformation over the past two decades, evolving from simple sequence alignment to sophisticated artificial intelligence approaches. Formally, we can highlight four generations of viral identification tools. The first generation emerged in the 1990s based on sequence alignment approaches, with BLAST as the ubiquitous tool that was used to compare sequences against reference databases but was limited to detecting related viruses [13,14]. The second generation emerged in the 2010s based on statistical models and hidden Markov models (HMMs), with VirSorter (2015) as a key tool that detected viral “hallmark” genes and analyzed k-mer frequencies beyond simple sequence similarity [15]. This probabilistic approach combined multiple genomic features like gene density and strand bias to distinguish viral from cellular DNA with improved sensitivity. The late 2010s marked the emergence of machine learning-based approaches for viral identification, with VirFinder (2017) using logistic regression to identify viral sequences based on k-mer frequencies [16]. This was followed by DeepVirFinder (2020), which pioneered deep learning using convolutional neural networks to recognize complex patterns without reference databases [17]. The current frontier encompasses two main approaches: hybrid methodologies that combine multiple detection strategies, exemplified by VIBRANT (2020) which integrates neural networks with HMM-based annotation to maximize sensitivity and specificity [18], and emerging large language model (LLM) approaches like ViraLM (2024) [19] that leverage pre-trained genome foundation models such as DNABERT-2 to achieve enhanced detection capabilities, particularly for short viral contigs that challenge traditional methods. These approaches were further advanced in the last few years by the development of integrated identification and assembly tools that use HMM to grab conserved motifs of distantly related viruses from large sequencing datasets and subsequent assembly to enhance the contigs into near-full genomes. In this systematic review, we summarize tools and approaches for virus discovery, tracing their development from simple sequence-based methods to sophisticated artificial intelligence systems.

## 2. Materials and Methods

We conducted a comprehensive literature search using PubMed (National Center for Biotechnology Information, Bethesda, MD, USA) [20] as the primary database in July 2025 to systematically identify computational tools developed for the discovery of viral sequences. We included tools that can at minimum distinguish viral from non-viral sequences, while also offering differing capabilities for taxonomic resolution where applicable. No restrictions were imposed on the year of publication, resulting in coverage of articles from 1990 through 2025. The search queries were designed to cover a wide range of virus detection tools and classification studies, using keywords such as “viral” and “metagenomics”, and “computational tool”.

In addition to a direct database search, a snowballing approach was implemented to broaden the scope. Relevant tools referenced in both reviews and primary research articles were incorporated, particularly when they were not indexed through our initial PubMed search. This strategy ensured a more exhaustive and representative selection of computational resources pertinent to viral taxonomy annotation.

Given the breadth and diversity of available tools, we focused our analysis on categorizing methods according to their underlying viral detection strategies. Recognizing that many tools implement hybrid approaches, we assigned each tool to the group corresponding to the methodological paradigm that is most prominent and foundational within its workflow. Consequently, we distinguished four principal approaches: alignment-, profile hidden Markov-, machine learning, and k-mer based approaches. In addition, foldome-based strategies, which rely on protein structural folds to infer evolutionary relationships, have recently emerged. While these methods hold promise for viral identification, and especially for further taxonomic assignment, they fall outside the direct scope of this review and are therefore not described in detail here. Some tools discussed in this review incorporate elements of such structural approaches, and the underlying algorithms are typically trained on either reference genomes or amino acid sequences. In most cases, protein data are used for the training process.

Inclusion criteria:

Peer-reviewed original research articles describing the development of a computational tool or pipeline for the identification and taxonomic annotation of viral sequences from eukaryotic, prokaryotic viruses, and proviruses.

Articles were required to provide full-text access with open code.

Relevance of the tool to the classification of viral sequences in the context of this review.

The exclusion criteria are shown in Figure 1. All articles that met the inclusion criteria and did not conflict with the exclusion criteria were reviewed in full text.

The tools in this review are suitable for metagenomic and virome studies.

Registration number 10.17605/OSF.IO/MKPFT (Center for Open Science, Charlottesville, VA, USA) [21].

**Figure 1 viruses-17-01538-f001:**
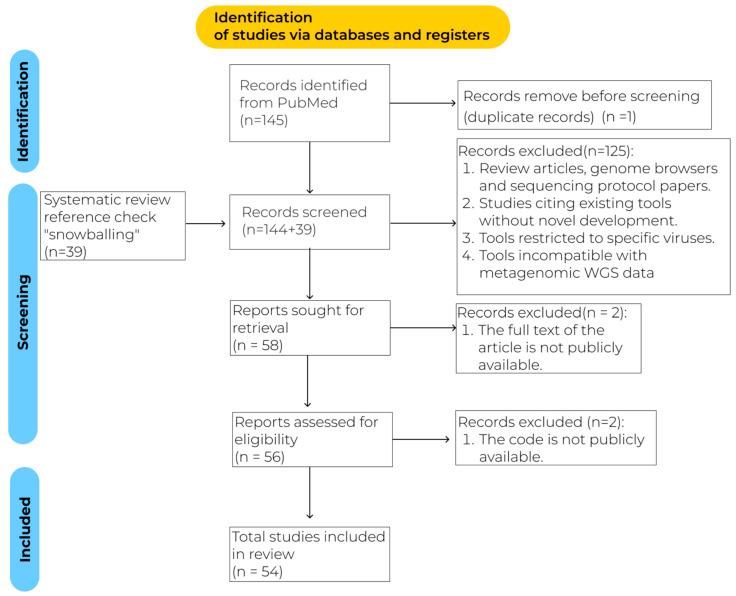
PRISMA flowchart.

## 3. Results

Based on the literature, the instruments have been categorized into several groups, as illustrated in Figure 2. The evolution of the instruments was illustrated in the timeline.

### 3.1. Alignment-Based Approaches for Virus Sequence Identification

Sequence alignment-based methods represent the classical computational approach for identifying viral sequences through systematic comparison with reference databases. These methods operate on the principle that evolutionarily related sequences retain detectable similarity despite accumulating mutations over time. The approach involves aligning unknown query sequences against known viral genomes or proteins to identify regions of similarity that indicate shared ancestry [22]. The core workflow consists of three stages: alignment computation [18,23], statistical evaluation, and taxonomic inference. Statistical frameworks are used to evaluate alignment significance using metrics like E-values, bit scores, and percentage identity. E-values (Expect values) represent the number of alignments with similar or better scores expected to occur by chance in a database search—lower E-values indicate more significant matches, with values below 0.001 typically considered biologically relevant [14]. Bit scores provide a normalized measure of alignment quality independent of database size, calculated from the raw alignment score using log-odds substitution matrices, where higher values indicate better alignments and scores above 50 generally suggest homology [24]. Percentage identity measures the proportion of exact matches between aligned positions, offering an intuitive metric of sequence similarity, though its interpretation varies by sequence length and type [14]. Finally, taxonomic assignment occurs through several approaches, most commonly using the closest matching reference sequences and genetic distances for direct classification within established viral taxonomy, or through the lowest common ancestor (LCA) method, which analyzes the top n hits to determine the deepest shared taxonomic level, providing more robust classification and reducing misclassification errors.

#### 3.1.1. Pairwise Alignment Methods

Pairwise sequence alignment is a fundamental method in bioinformatics that compares two biological sequences-DNA, RNA, or protein-to identify regions of similarity. This process involves finding the optimal alignment that maximizes similarity by introducing gaps (insertions or deletions) where necessary. Pairwise alignments can be either local or global: local alignment identifies and aligns the most similar subsequences within two sequences, even when the overall sequences differ significantly, with the Smith–Waterman algorithm serving as a widely adopted dynamic programming approach for this purpose. In contrast, global alignment attempts to align entire sequences end-to-end, matching every character, often with gaps; the Needleman–Wunsch algorithm is the classical method employed in global alignment [21].

Pairwise sequence comparison represents a fundamental pillar of viral taxonomic classification, as recognized by the ICTV [25,26]. These comparative methodologies analyze viral genomic sequences to quantify similarity thresholds, thereby defining the demarcation criteria that distinguish between different taxonomic ranks.

BLAST (Table 1) is a widely used local alignment search tool that identifies high-scoring local similarities between biological sequences [14]. Using weight matrices (such as PAM-120), it scores residue pairs, summing scores over aligned segments to find maximal segment pairs (MSPs). It efficiently searches large databases by indexing short k-mer words from the query and extending hits that exceed defined score thresholds, balancing sensitivity and speed. Tools implementing this approach can process both nucleotide sequences (BLASTn) and translated amino acid sequences (BLASTp), with protein-level comparisons offering enhanced sensitivity for divergent viruses.

The NCBI Virus portal [27] provides a specialized BLAST interface optimized for viral sequences, integrating results with curated viral metadata. MegaBLAST is an optimized version of the BLAST algorithm designed for the rapid detection of exact matches in large genomic databases [28]. It employs a database index containing compressed sequences and positional information on k-mers. MetaPhinder builds upon BLAST by utilizing metrics such as average nucleotide identity (%ANI) and sequence coverage relative to the database, evaluating the overall similarity across multiple matches rather than relying solely on the highest-scoring alignment [29].

PASC pioneered another approach at NCBI, comparing each new viral sequence against reference genomes using both local (BLAST) and global (Needleman–Wunsch) alignment algorithms. However, PASC shows limitations with highly diverse virus families or those exhibiting significant genome length variation [30]. SDT employs Needleman–Wunsch alignment algorithms, implemented through MUSCLE, ClustalW, or MAFFT, to perform pairwise sequence alignments. SDT can produce publication–quality pairwise identity plots and color-coded distance matrices to further aid the classification of sequences according to ICTV-approved taxonomic demarcation criteria [31]. VICTOR specializes in prokaryotic virus classification using genome BLAST distance phylogeny (GBDP) methodology, which employs modified BLAST comparisons across entire genomes rather than individual genes [32]. VIRIDIC has become a widely used tool for bacteriophage classification, calculating pairwise intergenomic similarities and performing hierarchical clustering with intuitive heatmap visualization [33]. The tool implements genus-level demarcation using 70% nucleotide similarity thresholds aligned with ICTV recommendations, though species-level boundaries vary by viral family (typically 95% for many bacteriophages). VIRIDIC processes multiple viral genomes simultaneously, generating color-coded matrices that reveal phylogenetic relationships and facilitate the identification of novel viral taxa. vConTACT [33,34,35] versions 1 and 2 utilize the Markov cluster algorithm (MCL) methodology to generate protein clusters (PC) based on BLASTP results. The similarity between genomes is estimated through the hypergeometric distribution, which calculates the probability that two genomes will randomly share n PCs, given the total number of PCs [31].

#### 3.1.2. Multiple Sequence Alignment Methods

While pairwise alignments are often sufficient for initial sequence identification, comprehensive virus analysis relies upon a multiple sequence alignment (MSA) to elucidate the relationships between multiple viruses with varying relationship degrees. This approach is fundamental in bioinformatics applications, such as reconstructing evolutionary histories and identifying functionally important motifs. However, the complexity of aligning multiple sequences presents significant computational challenges. To address these, various algorithms have been developed, including progressive methods that build alignments stepwise from pairwise comparisons, as well as iterative refinement techniques and other advanced strategies that enhance alignment accuracy and scalability.

PSI-BLAST [24] is an iterative amino acid sequence search algorithm that improves sensitivity by constructing and refining a position-specific scoring matrix (PSSM). Initially, it performs a standard BLAST search to identify significant matches, which are used to generate a multiple sequence alignment. From this alignment, a PSSM is derived, capturing position-specific amino acid conservation. The process iterates, updating the PSSM with new significant hits, improving detection of distant homologs. Iterations continue until no new significant sequences are found or a set maximum number of rounds is reached.

The standard tools for performing MSA are MAFFT [36,37], MUSCLE [38], ClustalW [39], ClustalX [39], and Clustal Omega [40,41]. ClustalW (and ClustalX, its slightly improved variant with a graphical interface) is a multiple sequence alignment algorithm that does progressive alignment, prioritizing sequences based on similarity scores and combining pairwise and global alignment approaches. Clustal Omega enhances scalability with the mBed embedding method for fast guide tree construction and aligns sequences using profile hidden Markov models combined with external profile alignment and iterative refinement for improved accuracy. MAFFT accelerates multiple sequence alignment by using the fast Fourier transform (FFT) to quickly identify homologous segments based on amino acid properties before applying dynamic programming. MUSCLE employs a three-stage progressive alignment process that starts with k-mer distance clustering, refines the guide tree with Kimura-corrected distances, and iteratively improves the alignment using a log-expectation scoring function. VIRULIGN [42] builds codon-correct alignments by comparing each target sequence to a reference using Needleman–Wunsch alignment of amino acid translations. It detects and corrects frame-shifts by adjusting problematic gaps, repeating until no frame-shifts remain or a limit is reached. VIRULIGN is designed to handle large sequence datasets; its main focus is on generating codon-correct multiple sequence alignments that prevent frameshifts, making it ideal for coding region alignments. In contrast, ViralMSA [43] is primarily developed for the rapid alignment of complete viral genomes in real time. ViralMSA is a flexible, cross-platform tool that rapidly aligns viral genomes by mapping sequences to a reference genome using Minimap2 by default, efficiently generating multiple sequence alignments while discarding insertions relative to the reference to focus on informative variations. It supports various mappers such as STAR [44], Bowtie 2 [45], and HISAT2 [46]. MACSE extends the classical Needleman–Wunsch algorithm to handle protein-coding nucleotide sequences with frameshifts and stop codons by considering 15 possible moves during alignment [47]. The alignment cost includes penalties for amino acid substitution, opening and extending gaps, as well as special high penalties for frameshifts and stop codons, which help maintain the correct codon structure in the alignment. For multiple sequence alignment, MACSE employs a progressive strategy using nucleotide k-mer frequencies for similarity estimation, constructing a guide tree via UPGMA, and aligning sequences and profiles with a pessimistic gap counting approach to accurately handle insertions and deletions. TranslatorX performs multiple sequence alignment of nucleotide sequences by translating them into amino acid sequences, aligning these amino acid sequences using established algorithms, and then back-translating the alignment to nucleotides while preserving codon structure [48]. To enhance alignment quality, TranslatorX employs a cleaning procedure based on the analysis of amino acid alignments using the GBlocks tool. This approach enables the removal of ambiguous regions from the nucleotide alignment while retaining informative sites and maintaining positional homology.

Virus taxonomic classification in GLUE is structured around an evolutionary hierarchy called an “alignment tree,” which organizes virus sequences into clades and clade categories reflecting their evolutionary relationships [49]. The alignment tree links parent and child clades through their reference sequences, ensuring coherent evolutionary representation. GLUE integrates MAFFT and BLAST+ for alignment. Sequence-to-clade assignment is performed using a Maximum Likelihood Clade Assignment (MLCA) algorithm, which places query sequences onto a fixed reference phylogenetic tree using the RAxML Evolutionary Placement Algorithm (EPA). Based on evolutionary distances to neighboring reference sequences already assigned to clades, MLCA calculates the likelihood of membership in each clade and assigns the query sequence to the most probable one above a confidence threshold.

#### 3.1.3. Rapid Similarity Estimation Methods

Traditional alignment-based approaches are computationally intensive, particularly for large-scale genomic datasets. To address this limitation, novel algorithms have been developed to minimize computational overhead while maintaining accuracy. Among these, MashMap stands out as an efficient method for estimating sequence similarity between reads and reference genomes. MashMap [49,50] approximates the Jaccard coefficient using a combination of MinHash sketching and winnowing techniques, bypassing the need for exhaustive alignment. The algorithm operates by indexing representative minimizers (k-mers) from both query and reference sequences, enabling rapid candidate region identification through a two-stage filtering process: initial filtering based on shared minimizer density to exclude non-homologous regions and precise similarity calculation using ordered data structures for retained candidates. This strategy efficiently narrows down potential matches, significantly reducing runtime while preserving high accuracy compared to conventional alignment methods. As a result, MashMap enables scalable and sensitive mapping, making it particularly suitable for large genomic datasets. MashMap’s underlying principles have been leveraged in FastANI [51], a tool designed for rapid Average Nucleotide Identity (ANI) computation. ANI serves as a key metric for quantifying nucleotide-level similarity between genomes, originally developed for bacterial and archaeal classification but now increasingly applied to viral genomes. FastANI exhibits high computational efficiency, processing thousands of genome pairs per minute, which makes it invaluable for large-scale genome clustering.

Vclust is a scalable viral genome analysis workflow that integrates k-mer-based similarity estimation (Kmer-db 2), precise pairwise sequence alignment using Lempel–Ziv parsing (LZ-ANI), and flexible clustering algorithms (Clusty) to efficiently process and classify millions of viral genomes [52].

Alignment-based methods remain the gold standard in virus identification. At some point, they are used in any such task. They are attractive because they are straightforward and relatively easy to set up. However, their main limitations include high calculation costs and low sensitivity to divergent viruses. Therefore, they may not be the best first-line solution for analyzing NGS data, and they are poorly suited for analyzing the “dark matter” of sequencing data, which might include genomes of highly divergent viruses. It should also be noted that, due to high genetic diversity and common indels (which often have the same position in the genome, but are not homologous), alignment of complete virus genomes at a family level cannot be fully automated. Key information, such as Methodology, Database source, Viral Specialization, Citation index (CI), and Limitations for alignment-based approaches, is presented in Table 1. While the citation index may indicate relative relevance of methods, it is important to consider that universal methods, such as BLAST, are used in many fields of biology.

### 3.2. Profile Hidden Markov Models Methods

Profile Hidden Markov Models (profile HMMs) are probabilistic frameworks that model the sequence variation within a family of related sequences (Table 2). Constructed from multiple sequence alignments (MSAs), profile HMMs capture patterns of conservation, insertions, and deletions characteristic of the sequence family. Compared to traditional similarity search methods such as BLAST, profile HMMs offer greater sensitivity, particularly in detecting distant homologs. As a distinct and foundational class of computational tools, profile HMMs have made invaluable contributions to the advancement of computational biology and molecular sequence analysis. HMMER is a widely adopted software implementation of profile HMMs for biological sequence analysis [53].

To build reliable profile HMMs, the input multiple sequence alignments (MSAs) must be carefully prepared. When training data includes many closely related sequences (e.g., many similar viruses from one species), this can bias the model by over-representing certain patterns. Sequence weighting schemes assign lower weights to redundant, closely related sequences and higher weights to more unique or divergent sequences. Dirichlet mixture priors incorporate knowledge about amino acid substitution patterns, improving the estimation of model parameters when training data is limited. The calibration process uses simulations to establish statistical parameters, enabling accurate E-value calculations essential for controlling false discovery rates in large-scale searches [54,55].

Modern viral identification tools often integrate profile HMM strategies within comprehensive bioinformatic pipelines to enhance sensitivity and specificity. VirSorter predicts viral sequences in complete or fragmented genome sequence data from bacteria and archaea by computing multiple statistically modeled metrics across sliding gene windows, including viral hallmark gene presence, viral-like gene enrichment, and various depletion or enrichment patterns in gene features [15]. It integrates hmm search with blastp search to sensitively detect viral hallmark genes and protein domains by comparing predicted proteins against curated viral HMM profile databases, which enhances its ability to find distant homologs and improve annotation accuracy. Detected regions are classified into three confidence categories based on combined metric significance and further refined iteratively by incorporating newly identified viral genes into reference databases. ViralRecall utilizes a nonredundant database of nucleo-cytoplasmic large DNA viruses (NCLDVs) genomes and constructs HMM profiles from clustering viral orthologous groups to identify viral sequences [56]. It calculates normalized scores from HMMER3 searches against giant virus orthologous groups (GVOGs) and Pfam databases to distinguish NCLDV-specific signatures, reducing false positives from related viral groups. Cenote-Taker2 is a comprehensive virus discovery and annotation pipeline that integrates BLAST-based and HMM-based methodologies [57]. The pipeline systematically annotates candidate tRNA genes and infers viral taxonomy through BLASTX comparisons against a curated database of viral sequences. ORF prediction is dynamically tailored according to inferred taxonomy, employing PHANOTATE for putative bacteriophages and Prodigal for all other viruses. Subsequent functional annotation of predicted ORFs is conducted via a rigorous, multi-tiered approach leveraging HMMER, RPS-BLAST, and HHblits to detect remote homologs against carefully curated protein domain repositories, ensuring precise and sensitive characterization of viral gene content. Phage_Finder combines protein homology, domain profiles, gene annotations, and integration site analysis to identify functional prophage regions in bacterial genomes [13]. Starting from windows exceeding a defined hit threshold, it expands candidate regions gene-by-gene based on HMM profile hits, BLASTP matches, presence of tRNA/tmRNA genes, and known phage annotations.

Phigaro [58] is a computational tool for prophage detection that leverages gene prediction (Prodigal) and phage-specific domain annotation (HMMER3/pVOGs) to score each gene based on curated “white-list” (prophage-enriched) and “black-list” (non-prophage-associated) profile hidden Markov models, with positive, neutral, or penalized weights assigned accordingly. These gene-level scores are integrated across genomic neighborhoods using the middle of the sliding window to smooth prophage likelihood estimates, and further refined by incorporating local GC content deviation from the host genome to improve boundary resolution, except when operating in GC-independent mode. It also produces dynamic annotated ‘prophage genome maps’ and marks possible transposon insertion spots inside prophages.

The viral RNA-dependent RNA polymerase (RdRp or replicase) is the most conserved protein in RNA viruses. Consequently, the high conservation of RdRp makes it a good phylogenetic marker for RNA viruses [59]. The palmdb is a curated database of viral polymerase palmprint sequences clustered into species-like OTUs at 90% amino acid identity, enabling standardized viral classification [60]. RdRp-scan constructs a comprehensive viral RdRp database by integrating sequences from palmdb and recent metagenomic studies [61]. The sequence is reduced via CD-HIT clustering. For each taxonomic cluster and unassigned group, multiple sequence alignments are generated using Clustal Omega, followed by manual curation. Hidden Markov Models are built from these alignments with HMMer3 under standard settings. Combining RdRp-specific HMMs and structural homology enables RdRp-scan to detect RdRp sequences sharing as little as 10% identity with known viruses. NeoRdRp2 [62] is a tool that works similarly. Sequences are clustered using CD-HIT at a 99% identity threshold to remove redundancy. Clusters with more than three sequences are aligned using MAFFT, followed by gap-based splitting with a custom script to refine conserved regions. Comparisons of eight different RdRp search tools showed that NeoRdRp2 exhibited balanced RdRp and nonspecific detection power.

Overall, HMM-based tools excel at the identification of unknown viruses. Their limitations include reliance on pre-identified profiles or laborious setup when aiming to use the most up-to-date genomic data and targeting of only the most conserved genome regions. Key information, such as Methodology, Database source, Viral Specialization, Citation index (CI), and Limitations for HMM-based approaches, is presented in Table 2. It is noteworthy that many HMM-based tools are not highly cited and are limited to narrow applications.

### 3.3. Machine-Learning-Based Approach

One advantage of machine learning methods is their ability to identify patterns that are challenging for humans to detect. Consequently, these methods can identify viral sequences that may not be present in databases despite the existence of underlying patterns. In principle, an ML method consists of a sequence representation and an analysis algorithm, which can be used in many combinations (Figure 3).

Early approaches to addressing the challenge of detecting viral sequences in metagenomic data were based on traditional machine learning methods, which, despite their simplicity, laid the foundation for more advanced algorithms. One of the first such tools was VirFinder [16], which implements logistic regression to predict viral and non-viral sequences. The underlying assumptions of VirFinder include that the distribution of k-mers (8-mers) in viral sequences correlates more strongly with their hosts than with random hosts, and that certain nucleotide combinations statistically differentiate viral from non-viral genome fragments. Beyond linear models, ensemble methods such as Random Forest (RF) have also been successfully applied for viral sequence detection in MARVEL [62], a tool developed specifically for identifying double-stranded DNA bacteriophages (dsDNA phages) of the order Caudovirales. The RF algorithm was also selected as ViraPipe’s primary classifier [63,64] using codon usage features (Relative Synonymous Codon Usage, RSCU) as a sequence feature. VirSorter2 constructs a comprehensive viral HMM database and utilizes it to derive 27 sequence-based features, which are then employed to train random forest classifiers for the accurate identification of viruses across diverse taxonomic groups [65]. PhiSpy [66] analyzes sliding windows of genes to compute features such as customized nucleotide skews, protein length differences, transcription strand consistency, and presence of unique phage-specific sequences, integrating homology information. Using a Random Forest classifier trained on related bacterial groups, it predicts prophage regions. VIBRANT [18] leverages a rigorously curated, nonredundant dataset of genomic fragments encompassing bacteria, archaea, plasmids, and viruses. These fragments undergo uniform gene prediction and protein annotation via multiple hidden Markov model (HMM) profile databases, from which 27 informative annotation-derived metrics are extracted. Employing these features, VIBRANT utilizes a multilayer perceptron neural network to robustly classify genomic fragments as viral or non-viral.

Convolutional neural networks (CNNs) were originally developed for image processing, but have since found wide application in the analysis of text data, including nucleotide or amino acid sequences. VirFinder developers created DeepVirFinder [17], a tool that uses a CNN trained on non-overlapping fragments of fixed lengths (150 bp, 300 bp, 500 bp, 1000 bp, and 3000 bp) of prokaryotic virus genomes and reference prokaryotic genomes. In comparative analyses, DeepVirFinder was shown to outperform VirFinder when identifying viral sequences in metagenomic data. PPR-Meta [17] is based on a Bi-path convolutional neural network (BiPathCNN) architecture that processes two distinct input matrices: a BOH matrix (representing base information useful for non-coding regions) and a COH matrix (representing codon information useful for coding regions). The CHEER [67] model is a hierarchical taxonomic classification framework designed for viral metagenomic reads, with a particular focus on RNA viruses. It organizes multiple CNN classifiers arranged in a tree structure from order to genus levels. Classification is performed top–down: first rejecting non-RNA viral reads, then classifying at the order level, followed by family-level classifiers within orders, and finally genus-level classifiers within families. Sequences are encoded via a k-mer embedding that captures k-mer co-occurrence and ordering information, improving performance. DeepMicroClass [68] employs a CNN with two input paths: a base-path encoding one-hot nucleotide-level information (A, C, G, T), including reverse complements, and a codon-path encoding codon-level information. VirDetect-AI [69] is a deep neural network based on the ResNet architecture combined with CNNs, designed for the classification of eukaryotic viral proteins. Amino acid sequences are preprocessed by fragmentation into overlapping k-mers and encoded using one-hot encoding into binary matrices.

Long short-term memory (LSTM) is a type of recurrent neural network (RNN) that can learn long-term dependencies thanks to the mechanism of “gates”. The main advantage of LSTM over other methods (such as CNN or Random Forest) is the ability to take into account the order of nucleotides/amino acids and identify complex, extended dependencies in the data. Seeker [70] is an LSTM-based deep learning model trained for the identification of bacteriophage genomes. HVSeeker [71] is a computational tool for sequence identification and classification that offers two LSTM-based models, DNA and protein-based. Similarly, Virtifier [72] uses LSTM to predict viral sequences using the Seq2Vec encoding method to represent the data.

DETIRE [73] employs a two-stage deep learning framework for virus identification from metagenomic data. Initially, a graph convolutional network (GCN) learns meaningful 30-dimensional embeddings of 3-mer nucleotide fragments. These embeddings serve as inputs to a hybrid classifier that integrates CNN to capture spatial features and bidirectional LSTM (BiLSTM) networks to model sequential dependencies. The combined features are jointly learned and classified via fully connected dense layers and a softmax output, enabling accurate discrimination between viral and non-viral sequences from fragmented metagenomic reads.

PhaGCN [74] employs a semi-supervised graph convolutional network framework for bacteriophage identification.

The tool geNomad [6] implements a hybrid approach for the identification of viruses, plasmids, and proviruses in metagenomic data by combining two methods: a sequence-based classifier and a gene-based classifier. The gene-based approach relies on the XGBoost algorithm, where each amino acid sequence is represented by a vector of 25 features that serve as input to the classifier. The sequence-based classifier applies a two-step model: first, a neural network encoder transforms nucleotide sequences into fixed-length vector representations [75]. IGLOO’s architecture is designed under the assumption that sequences belonging to the same class should have similar vector embeddings (i.e., cluster closely in feature space), while sequences from different classes should be well-separated. The resulting vector is then fed into a fully connected (dense) neural network that performs the final classification. The outputs of these two independent classifiers—gene-based and sequence-based classifiers—are integrated using a feedforward neural network to produce the final decision regarding the class membership of the analyzed sequence.

VirNucPro [76] employs dual embeddings: DNABERT_S for nucleotide sequences and ESM2-3B for amino acid sequences, producing contextual feature vectors.

GCNFrame [77] employs a novel representation of genomic sequences as gapped pattern graphs (GPGs), which encode both contiguous and non-contiguous k-mer patterns to capture genetic variations, including single-nucleotide polymorphisms (SNPs) and insertions/deletions. These graph-structured data are processed using a graph convolutional network based on the GraphSAGE framework, which learns high-quality, low-dimensional embeddings that encapsulate complex sequence features and variation patterns. Subsequently, these embeddings serve as inputs to fully connected MLPs for accurate classification of genomic attributes such as bacteriophage identification, lifestyle characterization, and host range prediction.

VirRep [78] is a hybrid language representation learning framework for identifying viruses in human gut metagenomic data. It combines a BERT-like semantic encoder that captures k-mer patterns with a BiLSTM alignment encoder that encodes sequence similarity to prokaryotic genomes. The tool processes sequences by segmenting them into 1 kb fragments, tokenizing into 7-mers, and generating probability scores through a siamese neural network analyzing both strands. VirRep uses a three-stage training approach: pre-training both encoders separately, fine-tuning each on specific tasks, then combining them with a binary classifier.

ViraLM [19] implements a foundation model-based approach for identifying viral sequences in metagenomic data by leveraging the pre-trained genome foundation model DNABERT-2 with masked language modeling (MLM). The architecture consists of a transformer attention block initialized with DNABERT-2’s parameters, followed by a binary classification layer for virus identification.

All the algorithms described above were trained either on reference genomes or amino acid sequences. Typically, protein data are used to complement the training process, and only one tool performs complete training solely on proteins. Before training, a data preprocessing stage is applied. During this stage, sequences may be segmented into k-mers or fragmented using various approaches such as non-overlapping sequences of different lengths, padding, or sliding window techniques. These fragments can then undergo different encoding processes to convert them into numerical representations suitable for machine learning models. Training can be performed directly on raw reads, which may be assembled and aligned against reference databases to improve classification accuracy. The scheme of training and preprocessing of algorithms is presented in Figure 3.

Key information, such as Methodology, Database source, Viral Specialization, Citation index (CI), and Limitations for ML-based approaches, is presented in Table 3. It is noteworthy that most of the ML-based tools have been developed for the detection of dsDNA viruses, primarily phages. Their general utility for RNA virus identification may be questioned because RNA viruses exhibit remarkable variation in genome composition, even at the family level, whereas most ML-based methods were designed to handle nucleic acid sequences. Also, the complexity of ML-based virus identification tools is rapidly increasing; however, there is limited evidence that this leads to a significant prediction quality gain. In addition, citation values indicate that many tools were not integrated into routine virus discovery.

### 3.4. K-Mer-Based Approach

K-Mer-based approaches have gained significant popularity due to their high computational speed. One of the most well-known algorithms in this category is Kraken2 [79]. At the core of Kraken lies a database containing records composed of a k-mer (31-mers by default) and the LCA of all organisms whose genomes include that k-mer. This design introduces a key limitation: Kraken2 can only identify viruses present in its reference database. As expected, k-mer-based methods often exhibit lower sensitivity and specificity when identifying species in complex metagenomic samples compared to full sequence alignment methods, but they are substantially faster [80]. DisCVR [81] uses the k-mers approach in which the sample reads are decomposed into k-mers and then matched against a virus k-mers database. KAnalyze [82] was chosen for integration into DisCVR because the k-mers it generates are sorted lexicographically, thus making the search for matches very efficient.

Another widely used tool for taxonomic classification is CLARK [83]. Unlike standard approaches, CLARK constructs its database exclusively from discriminative k-mers unique to each taxon, improving classification accuracy. However, traditional frequency-based k-mer representations do not capture the order of k-mers, potentially losing important structural information about sequences. To address this, recent studies have adopted techniques from natural language processing, such as Skip–Gram models and other embedding methods, which consider not only k-mer composition but also their contextual co-occurrence. These models learn which k-mers tend to appear together and map them into vector spaces, such that semantically related k-mers have similar embeddings, as demonstrated in tools like CHEER. We also find the use of k-mers to speed up alignment as implemented in Mashmap. Thus, the k-mer-based approach may be a stand-alone tool or integrated into various programs to optimize algorithm operation.

In metagenomic datasets, viral reads often constitute only a small fraction of total sequenced data, making their detection and analysis methodologically challenging. Genome assembly, a computationally intensive step, can be partially circumvented by pre-filtering reads likely to be viral before assembly and alignment stages. This pre-filtering reduces computational load and improves downstream analysis [81]. Moreover, k-mer-based filtering can complement assembly-based and homology-based methods, enhancing viral sequence identification as demonstrated by the AliMarko pipeline [84].

VISTA [85] employs an integrative approach combining pairwise sequence comparisons, k-mer profiling, and machine learning to achieve robust viral identification. The method begins with the extraction of k-mer profiles that encode both sequence composition and positional information derived from physicochemical properties of protein translations. Feature selection via chi-squared statistics and the extremely randomized trees (ERT) identified the most discriminative k-mer signatures for taxonomic classification. The optimal subset of features extracted from k-mer profiles was utilized as input to calculate pairwise measures of distance among all genome sequences. These distances are normalized and analyzed using Gaussian kernel density estimation to identify natural thresholds corresponding to taxonomic ranks (species, genus, family). Hierarchical clustering guided by these thresholds is evaluated against a known taxonomy to select optimal cutoffs. For an unknown viral genome, VISTA constructs its k-mer profile and computes distances to reference sequences. By comparing the minimum distance to the established taxonomic thresholds, the genome is assigned to an existing species if below the species cutoff, to a new species within a genus if below the genus cutoff, or to a different genus or family if above these thresholds.

Key information, such as Methodology, Database source, Viral specialization, Citation index (CI), and Limitations for k-mer-based approaches, is presented in Table 4.

We summarized key information (*n* = 19 parameters) for all 54 viral detection tools based on their original publications in Appendix A. The summary included details such as methodology, database source, input type, advantages, limitations, etc.

## 4. Discussion

Viruses comprise a highly diverse community. Many existing tools were trained on specific databases representing only particular groups, such as viruses infecting prokaryotes, eukaryotes, or RNA viruses. In this systematic review, we present a selection of tools that cover a broad spectrum of viruses, including bacteriophages, eukaryotic viruses, large DNA viruses, and others. Tools that focus exclusively on a single virus genus were excluded. When designing a study, it is crucial to carefully choose the appropriate tool based on the viral group under investigation.

Approaches for virus discovery vary widely, ranging from classical alignment-based methods to advanced alignment-free machine learning models. The complementary strengths of these methods suggest that optimal virus identification strategies should integrate multiple approaches. Alignment-based methods remain the gold standard for well-characterized viruses with close database homologs. HMM-based tools excel at detecting conserved protein domains across divergent sequences. K-Mer methods provide rapid initial screening and taxonomic classification of known viruses. Machine learning approaches may fill critical gaps by identifying novel viruses, handling fragmented sequences, and learning complex patterns invisible to traditional methods. As metagenomic datasets grow exponentially, hybrid pipelines combining k-mer-based pre-filtering with ML classification and targeted alignment verification will likely become standard practice for comprehensive virome characterization.

We also observe a growing trend toward hybrid methods that combine multiple strategies, with machine learning integrated with k-mer analysis becoming particularly prominent. These modern tools have evolved beyond simple frequency counting by incorporating sophisticated machine learning techniques to address the limitations of traditional k-mer approaches. While conventional methods like Kraken2 perform direct database matching of individual k-mers, newer hybrid tools such as CHEER and VirRep apply natural language processing techniques, including Skip–Gram models and BERT-based encoders, to learn k-mer co-occurrence patterns and contextual relationships. By treating DNA sequences analogously to text, where k-mers function as “words” with meaningful relationships, these methods capture not just k-mer presence but their typical co-occurrence patterns in viral versus non-viral sequences. Similarly, tools like geNomad combine machine learning with homology searches, integrating sequence-based neural network classifiers with gene-based XGBoost models to leverage both pattern recognition and similarity-based detection. This evolution toward hybrid approaches creates a new generation of tools that maintain the computational efficiency of traditional methods while incorporating sophisticated pattern recognition capabilities, effectively bridging the gap between fast but limited conventional approaches and computationally intensive pure deep learning models.

The Global Virome Project (GVP), which endeavors to characterize an estimated 1.67 million previously undescribed viral species with approximately 631,000–827,000 possessing zoonotic potential, has fundamentally underscored the imperative for sophisticated and reliable bioinformatics methodologies in viral taxonomic annotation [86]. Nevertheless, the computational landscape for viral classification is characterized by the continuous emergence of novel methodologies, complicating the determination of optimal approaches without standardized benchmarking protocols and evaluation frameworks [87]. This deficiency in systematic comparative assessment precludes definitive determination of tool performance characteristics and consequently impedes evidence-based methodology selection for viral detection applications.

While the present review does not aim to provide benchmarking assessments of the described tools, analysis of recent large-scale virome initiatives offers valuable insights into which computational approaches have demonstrated scalability and practical utility in extensive viral discovery efforts. However, choosing optimal pipelines depends on particular research tasks, as different studies require distinct analytical strategies. To illustrate this, we selected several representative studies that demonstrate remarkable diversity in analytical approaches. For instance, Zhang et al. (2025) developed a comprehensive workflow combining Kraken2 for host read filtering and Diamond [88] BlastX/BlastN searches against nr/nt databases, followed by MMseqs2 [89] clustering to identify viral operational taxonomic units from 1113 small mammals [90]. Similarly, Guo et al. (2022) constructed a specialized detection framework using custom databases of human, archaeal, bacterial, and vector sequences alongside NCBI viral references, employing Kraken2 for initial classification and BLAST for species-level annotation with stringent filtering criteria to analyze blood virome data from over 10,000 individuals [91]. Several large-scale gut virome studies have incorporated machine learning-based viral identification tools into their analytical pipelines. Nayfach et al. (2021) [92] utilized VirFinder’s k-mer frequency machine learning models to identify 189,680 DNA viral genomes from human gut metagenomes, while Zeng et al. (2024) [93] combined VirFinder with VIBRANT’s hybrid ML approach and VirSorter2’s automated classifiers to catalog 160,478 viral sequences from early-life gut samples. Similarly, Nishijima et al. (2022) [94] employed DeepVirFinder for viral-specific k-mer pattern detection in their analysis of 4198 Japanese individuals, and recent studies like Yan et al. (2025) [95] integrated DeepVirFinder with VIBRANT to construct the Chinese Gut Virus Catalog containing 426,496 viral sequences, while Galperina et al. (2025) [96] used DeepVirFinder and VirSorter2 alongside geNomad and PhaGCN to create the Aggregated Gut Viral Catalog with over 1 million dereplicated viral sequences.

A challenge in the identification of virus sequences, especially those of RNA viruses, comes from high genetic variability. Most RNA viruses accumulate about 10^−3^ substitutions/site/year. This renders nucleotide sequence-based approaches poorly usable for the identification of even moderately related viruses. Moreover, above a family level, only a few proteins (usually polymerase and protease) may be recognized by the most sensitive methods (usually HMM-based), while most of the genome cannot be identified at all. This challenge calls for integrated identification-assembly methods that first identify anchor sequences by HMM screening and then work over raw sequencing data to specifically extend virus contigs [97]. This approach allowed the identification of multiple novel viruses in publicly available read archives [98].

The field of viral taxonomy is rapidly evolving, driven by ongoing discoveries and the integration of novel genomic data. Maintaining up-to-date reference databases is critical for the development and accuracy of virus classification models. For instance, the ICTV reported a substantial increase in the number of recognized virus species, from 9110 species in 2020 to 16,215 species by 2024 [99]. Without regular updates to the underlying training databases, computational classification tools risk becoming outdated, potentially leading to misclassification or reduced sensitivity in recognizing emerging viral diversity. Therefore, ongoing curation and timely incorporation of the latest taxonomic releases are essential to sustain the performance and relevance of predictive models in viral genomics.

A significant limitation encountered in some studies is the lack of publicly accessible source code. Without open and reproducible code, the scientific community faces challenges in independently evaluating, validating, and benchmarking these computational methods, especially as they were trained at different times and on different datasets. This lack of transparency potentially undermines confidence in the forecasts of the tools and their integration into broader workflows. This issue reflects a wider reproducibility crisis in computational virology, where many published tools cannot be reliably replicated or compared due to unavailable code and undocumented parameters. Strengthening reproducibility is therefore essential to ensure scientific credibility and long-term usability of these resources. Therefore, to advance viral taxonomy computationally and ensure community trust, future contributions must prioritize transparency by providing fully accessible, well-documented code and a reproducible workflow that allows re-training. Such practices will facilitate rigorous peer assessment, foster methodological improvements, and enable the field to keep pace with the rapidly expanding virosphere.

The most popular database chosen for training and annotation is NCBI viral RefSeq, a comprehensive resource comprising over 6600 virus species with high-quality genome annotations as of July 2024 [100]. It integrates closely with the ICTV by adopting exemplar isolates designated as reference representatives for viral species, ensuring taxonomic consistency.

This work has systematically reviewed the most advanced tools in virus detection, highlighting the fundamental role that alignment, k-mer, profile HMMs, and machine learning approaches play in virology bioinformatics. Our systematic review has some limitations. The initial search was conducted only in PubMed. Although we applied the snowballing principle, certain studies may not have been captured. Furthermore, the search query itself may have limited the scope of the search. Finally, our review included only tools that had been published up to September 2025.

## 5. Summary

The choice of virus identification methods depends upon the object, because there are different levels of genome conservation among diverse virus families and at different taxonomic levels. Also, the goal defines the means, be it fast screening for known viruses in surveillance applications or deep exploration of the “dark matter” of sequencing experiments.

Alignment-based methods are optimal for well-characterized viruses requiring high-confidence identification, clinical diagnostics, and precise strain-level taxonomic assignment in small-scale analyses. They excel at validating novel virus claims and distinguishing closely related strains, but may not be the primary choice for large metagenomic datasets, novel virus discovery, and highly divergent sequences. The fundamental limitation is their inability to detect viruses without similar references in the database, restricting their utility to confirmation rather than discovery.

Rapid similarity estimation tools enable high-throughput classification of millions of viral genomes for ANI-based taxonomic assignment, outbreak surveillance, and building viral operational taxonomic units following ICTV and MIUViG standards. While excelling at dereplication and rapid genotype assignment in time-critical scenarios, they should be avoided for novel virus discovery, sequences with less than 70% ANI to references, and protein-level analysis.

Profile HMM approaches are particularly valuable for novel virus discovery through conserved protein domain detection, identifying divergent viral sequences, and specialized tasks such as prophage detection or RNA virus identification. They are especially useful when whole-genome similarity is too low for alignment methods, but should be avoided for very short contigs with high false positive risk, extremely large datasets with limited computational resources, and when database coverage is sparse.

K-Mer-based methods provide ultra-fast preliminary screening of massive metagenomic datasets and abundance estimation of known viruses when speed is paramount and computational resources are limited. However, they should be avoided for novel virus discovery, precise strain-level differentiation, and divergent sequences, as they are restricted to exact or near-exact k-mer matches against database content.

Machine learning methods enable database-independent novel virus discovery by integrating multiple genomic features to identify intrinsic viral sequence patterns, making them ideal for fragmented metagenomic assemblies and under-sampled viral diversity. They should be avoided for taxonomic assignment and in contexts requiring interpretable results, as their black box nature limits explainability.

The most effective approach involves layering methods strategically: fast screening with k-mer or machine learning, domain validation with profile HMMs (probably combined with re-assembling or contig extension), precise characterization with alignment-based or rapid similarity methods, and quality assessment with combination pipelines. Tool selection fundamentally depends on whether one is working with known viruses, favoring database-dependent methods, or pursuing novel discovery requiring database-independent approaches.

Using this information from the original publications, we created a roadmap to guide tool selection according to research goals and virus types presented in Figure 4.

## Figures and Tables

**Figure 2 viruses-17-01538-f002:**
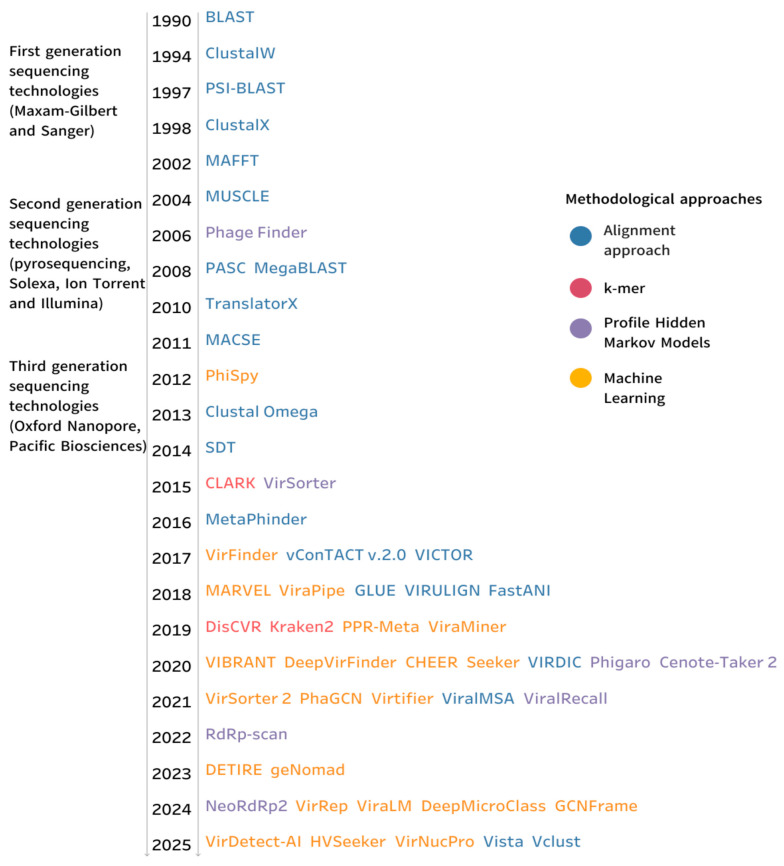
Evolution of viral detection tools and methodological approaches. The timeline illustrates the development of key computational tools for viral taxonomic annotation from 1990 to 2025. Each tool name is colored according to its underlying methodological approach.

**Figure 3 viruses-17-01538-f003:**
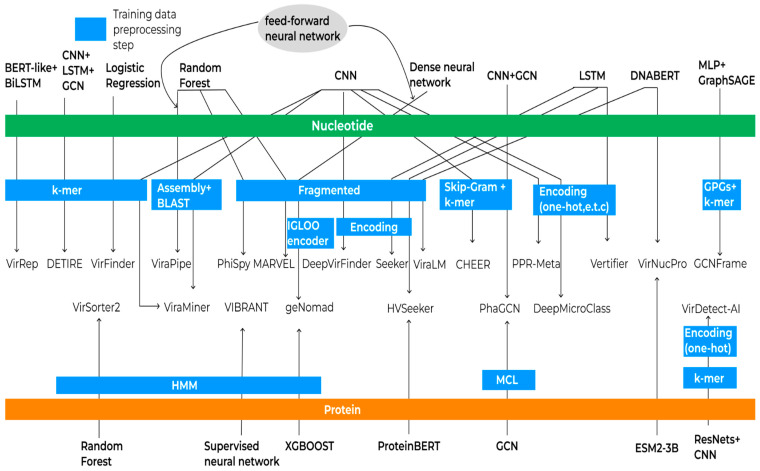
Schematic representation of machine learning algorithms and data processing workflows for virus identification. Green and orange bars indicate the type of training data utilized (nucleotide and amino acid sequences, respectively). Each arrow originates from a machine learning method, traverses relevant data preprocessing steps, and points to the corresponding analytical tool, illustrating the flow from data preparation to application in virus identification.

**Figure 4 viruses-17-01538-f004:**
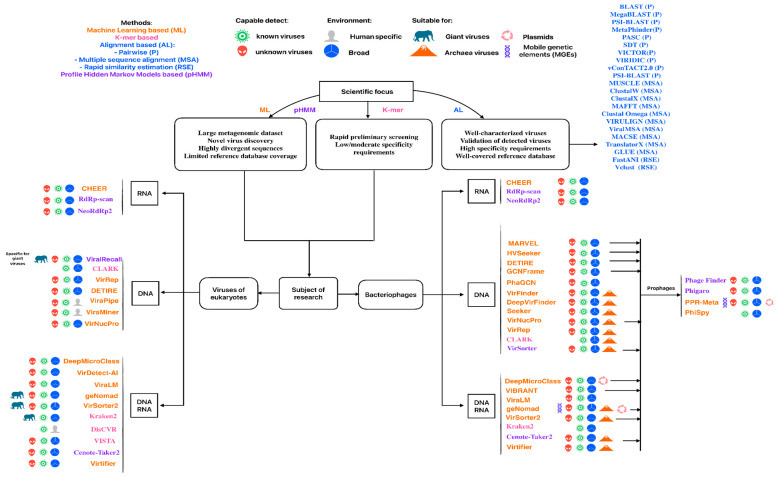
Schematic guide for selecting viral analysis tools according to specific research goals and virus types. Each tool name is colored according to its underlying methodological approach.

**Table 1 viruses-17-01538-t001:** Alignment-based tools for virus detection.

No.	Tool	Methodology	Database Source	Viral Specialization	CI	Limitations
**1**	BLAST	Pairwise alignment	NCBI RefSeq Viral	DNA viruses, RNA, eukaryotic viruses, phages, archaeal viruses, and protein queries across all viral taxa	120,903	- Slower than MegaBLAST for near-identical sequences - Requires pre-formatted databases - May miss extremely weak similarities - Computationally heavy for very large datasets
**2**	MegaBLAST	Pairwise alignment (fast)	NCBI RefSeq Viral	DNA viruses, eukaryotic viruses, phages, and archaeal viruses	1620	- Performance degrades with very long queries (>100 kb) or highly repetitive regions - Memory-intensive indexing - Limited sensitivity for highly divergent (low-identity < 70%) or RNA virus sequences
**3**	PSI-BLAST	Multiple sequence alignment (iterative)	NCBI RefSeq Viral	DNA viruses, RNA viruses, eukaryotic viruses, phages (bacteriophages), and archaeal viruses	89,349	- Deceptive alignments caused by highly biased amino acid regions, which can lead to a high false positive rate
**4**	PASC	Pairwise alignment	NCBI	DNA viruses, RNA viruses, eukaryotic viruses, phages, and archaeal viruses	196	- Some virus families show overlapping peaks, making demarcations difficult - Global alignment is less reliable for large or distantly related genomes
**5**	MetaPhinder	Pairwise alignment	NCBI RefSeq Viral, EMBL EBI, phagesdb	dsDNA bacteriophages and archaeal viruses	78	- Requires similarity to known phage genomes - Drop in accuracy for short contigs (<5 kb)
**5**	SDT	Pairwise alignment	ICTV	Plant ssDNA viruses and novel circular DNA viruses from metagenomic studies optimized for small circular DNA viruses, but applicable to any virus family with ICTV identity thresholds	1817	- Alignment quality affects results for highly divergent genomes
**6**	VICTOR	Pairwise alignment	INSDC	Prokaryotic (bacterial and archaeal) DNA viruses, mainly tailed dsDNA phages (order Caudovirales)	617	- Requires nearly complete genomes - Lower resolution at family rank - Computationally heavy for very large datasets
**7**	VIRDIC	Pairwise alignment	GenBank	Prokaryote-infecting dsDNA viral genomes (bacteriophages, archaeal viruses)	669	- BLASTN sensitivity limit (65%) can miss distant relationships - Draft/scaffold genomes with gaps cause underestimation - Repetitive regions may bias results
**8**	vConTACT v.2.0	Pairwise alignment	NCBI RefSeq Viral	dsDNA viruses infecting Bacteria and Archaea	295	- Requires rebuilding reference networks when adding new data - Struggles with short contigs, overlapping genomes, and highly mosaic viruses
**9**	ViralMSA	Multiple sequence alignment	GenBank	RNA, DNA, eukaryotic, and phage viruses	86	- Discards insertions relative to the reference genome - Not suitable for viruses lacking a reliable reference genome - Dependent on external mapper performance
**10**	VIRULIGN	Multiple sequence alignment	-	RNA and DNA viruses, including eukaryotic viruses, phages	73	- Requires a reliable reference sequence and annotation - Discards sequences with excessive frame-shifts
**11**	MAFFT	Multiple sequence alignment	-	RNA viruses, DNA viruses, phages (bacteriophages), archaeal viruses, and eukaryotic viral genes	17,706	- Slightly less accurate than structural aligners for very distantly related proteins - Requires parameter tuning for low-similarity sequences
**12**	ClustalW	Multiple sequence alignment	-	DNA, RNA, phage, archaeal, and eukaryotic viruses	75,099	- Early misalignments in progressive steps are not re-optimized (local-minimum problem) - Accuracy declines for extremely divergent sequences
**13**	ClustalX	Multiple sequence alignment	-	DNA, RNA, phage, archaeal, and eukaryotic viruses	3262	- Alignment quality can be sensitive to the initial choice of parameters
**14**	Clustal Omega	Multiple sequence alignment	-	DNA, RNA, phage, archaeal, and eukaryotic viruses	1458	- Initial misalignments may remain uncorrected in the final alignment - Uses the MAC algorithm for profile alignment (demands high memory for large datasets)
**15**	GLUE	Multiple sequence alignment	GenBank	Hepatitis C virus (HCV), Rabies virus, Bluetongue virus, Hepatitis B virus, Circoviridae, Parvoviridae, Flaviviridae, Retroviridae	118	- Assumes a single evolutionary history per alignment tree - Insertions relative to the reference lose homology information
**16**	MUSCLE	Multiple sequence alignment	-	DNA viruses, RNA viruses, bacteriophages, archaeal viruses, and eukaryotic viruses	50,759	- Performance drops for extremely divergent sequences (<15% identity) - Progressive alignment errors are irreversible (no global re-optimization)
**17**	MACSE	Multiple sequence alignment	-	RNA Viruses (ssRNA, dsRNA), Plant RNA Viruses (Tobamovirus, Potyvirus), Archaeal and Eukaryotic DNA Viruses (NCLDVs, Adenoviridae)	693	- Slower than standard tools like MUSCLE or TranslatorX due to algorithm complexity - Ignores certain frameshift event types, leading to approximate solutions - Requires parameter tuning for optimal performance
**18**	TranslatorX	Multiple sequence alignment	-	RNA viruses, DNA viruses, phages (bacteriophages), archaeal viruses, and eukaryotic viral genes, Plant and archaeal viruses	1505	- Cannot accommodate frameshifts - Cannot explicitly handle frameshifts or true pseudogene sequences - Relies on external aligners for performance and accuracy - Limited performance on extremely fragmented contigs (<100 bp)
**19**	FastANI	Rapid similarity estimation (ANI)	-	large dsDNA viruses (>200 kbp, NCLDVs, giant phages, poxviruses)	4694	- Performance drops significantly below ~80% ANI for divergent sequences - Sensitive to poor assembly - Computationally intensive for large databases
**20**	Vclust	Rapid similarity estimation (ANI) + k-mer + clustering	RefSeq, GenBank, IMG/VR v4.1, Kmer-db2	DNA viruses, RNA viruses, bacteriophages, archaeal viruses, and eukaryotic viruses	8	- Performance may decrease with large datasets of highly similar genomes

**Table 2 viruses-17-01538-t002:** HMM-based tools for virus detection.

No.	Tool	Methodology	Database Source	Viral Specialization	CI	Limitations
**1**	VirSorter	Profile hidden Markov models	RefSeq Virus genomes, data from public studies	Bacterial and archaeal dsDNA viruses (Caudovirales and non-Caudovirales); detects integrated prophages and free lytic viruses	1149	- Limited detection of eukaryotic viruses (database bias) - Inefficient for short contigs (<3 kb) or non-assembled reads - May include false positives (genomic islands, MGEs) - Does not analyze integrase/att sites, less accurate for complete prophage boundaries
**2**	Cenote-Taker 2	Profile hidden Markov models + BLAST	Hallmark gene HMM (Hmmer) database, CDD, Pfam, PDB, GenBank, RefSeq	All virus classes with DNA or RNA genomes, Prokaryotic, Prophages, Archaeal, Eukaryotic	141	- Does not allow automatic identification of ribosomal frameshifts and intron-containing genes - Performance depends on the presence of recognizable hallmark genes - May miss minimal or highly degraded viruses lacking core proteins
**3**	Phigaro	Profile hidden Markov models	pVOGs	Prophage	186	- Slightly lower Jaccard index compared to PHASTER and VirSorter - Sensitivity depends on the selected detection mode
**4**	Phage Finder	HMM + BLAST + tRNAscan-SE + Aragorn + fasta33 + MUMMER	NCBI or WU BLASTP data, HMMSEARCH data, tRNAscan-SE data, Aragorn data, Phage_Finder information file, GenBank	Prophage	368	- Can skip “tandem” (piggy-back) prophages integrated into a single genomic site - Many putative prophage regions lack core HMM matches (large terminase, portal, major capsid), which may cause missed detections under strict mode settings
**5**	RdRp-scan	Hidden Markov Models	NCBI Riboviria’s proteins, PALMdb, data from public studies	Primarily eukaryotic RNA viruses (+ssRNA, -ssRNA, dsRNA), with partial coverage of prokaryotic RNA phages	92	- Short or fragmented ORFs (<200 aa) are often missed - Requires manual validation for some motifs and structural matches - Performance depends on database updates (must be regularly maintained) - May miss extremely divergent RdRps that fall outside of existing HMM or structural models
**6**	NeoRdRp2	Hidden Markov Models	PALMdb, NCBI RNA Virus database, UniProtKB, GenBank, data from public studies	RNA viruses (primarily eukaryotic), including prokaryotic RNA phages such as Leviviricetes	5	- May split core RdRp motifs (A-C) during HMM construction - False positives in a small fraction (188/564k non-RdRp hits) - Extremely divergent RdRps may still escape detection
**7**	ViralRecall	Profile hidden Markov models	Data from public studies	nucleo-cytoplasmic large DNA viruses (NCLDV)	80	- Potential for false positives on short fragments - Time-consuming and not feasible for large datasets

**Table 3 viruses-17-01538-t003:** ML-based tools for virus detection.

No.	Tool	Methodology	Database Source	Viral Specialization	CI	Limitations
1	VirSorter 2	ML (RF + hidden Markov models)	JGI Earth’s virome project, Xfams, NCBI RefSeq genomes, including archaea, bacteria, protists, fungi, and viruses	dsDNA phage, nucleo-cytoplasmic large DNA viruses (NCLDV), RNA viruses, ssDNA viruses, prophages	1151	- Performance drops on short contigs (<3 kb) - Some false positives with plasmids and eukaryotic mobile elements - High computational demand due to multi-classifier ensemble - RNA virus predictions require high-quality metatranscriptomic assemblies
2	VirFinder	ML (LR + k-mer)	RefSeq	Prokaryotic dsDNA viruses (bacteriophages and archaeal viruses)	653	- Lower performance for archaeal viruses
3	MARVEL	ML (RF)	RefSeq	dsDNA bacteriophages	186	- Dependent on upstream binning quality (chimeric bins cause errors)
4	ViraPipe	ML (RF)+ FFNN	NCBI GenBank, 19 different NGS experiments	Human DNA viruses	56	- Lower recall (few viral contigs detected at high precision) - Performance declines on very short or noisy contigs - Requires coding regions
5	VIBRANT	ML (NN)	RefSeq and Genbank, KEGG KoFam, Pfam (v32), and Virus Orthologous Groups (VOG)	Bacterial and archaeal dsDNA, ssDNA, and RNA viruses, capable of integrated provirus detection (Prophages)	951	- Lower recovery for very short (<1 kb) contigs - Depends on high-quality ORF prediction - Potential minor bias toward NCBI-trained proteins
6	PhiSpy	ML (RF)	Phantome server	Integrated bacterial dsDNA prophages	561	- Less effective for fragmented assemblies or metagenomic datasets - Misses very small or highly degraded prophages - Requires annotated ORFs for accurate detection.
7	DeepVirFinder	ML (CNN)	RefSeq	dsDNA viruses infecting bacteria and archaea	594	- Possible misclassification of eukaryotic contamination - Slightly less accurate for extremely short fragments (<150 bp)
8	PPR-Meta	ML (BiPathCNN)	RefSeq	Phages and plasmids (bacterial and archaeal MGEs, temperate and lytic)	181	- Slightly lower plasmid classification accuracy - Limited to prokaryotic MGEs (mobile genetic elements)
9	ViraMiner	ML (CNN)	Data from public studies	Human DNA viruses	152	- Lower recall for rare viral classes - Optimized for human-associated datasets - Less effective for environmental metagenomes
10	CHEER	ML(CNN) + k-mer	RefSeq	RNA viruses (human, animal, bacterial, archaeal, and environmental RNA viromes)	65	- Trained only on RefSeq viral genomes (may miss rare or novel taxa)
11	DeepMicroClass	ML (diPathCN)	NCBI Genome database, Kaiju, the PR2 database, MMETSP project, PLSDB, Virus–Host DB	Broad coverage of prokaryotic (including archaeal) and eukaryotic viruses—encompassing dsDNA, ssDNA, dsRNA, ssRNA viral genomes, plasmids, and prophages	13	- Possible misclassification between plasmids and prokaryotic chromosomes - Slightly reduced sensitivity for integrated proviruses - Not specialized for short reads (<500 bp)
12	VirDetect-AI	ML (CNN + RNN)	Virus Protein Database NCBI,	Eukaryotic DNA/RNA viruses	2	- Limited performance for short proteins (<300 aa) - Slightly reduced sensitivity for rare viral families (e.g., Orthoherpesviridae, Retroviridae)
13	Seeker	ML (LSTM)	RefSeq	Bacteriophages dsDNA (Caudovirales and unclassified tailed phages)	131	- Less accurate for very short sequences (<1 kbp) - May produce false positives for sequences with phage-like motifs but non-viral origin - Performance sensitive to sequence length distribution and GC-content biases
14	HVSeeker	ML (LSTM + ProteinBER)	NCBI, Integrated Microbial Genomes, Microbiomes–Viruses (IMGVR)	Bacteriophage dsDNA and bacterial host sequences	1	- RNN-based (LSTM) models can overfit long sequences (>1.5 kb) - Performance slightly lower for extremely short (<200 bp) fragments
15	Virtifier	ML (LSTM)	Refeq viral	Prokaryotic and eukaryotic DNA viruses	48	- Reduced accuracy when training/testing lengths mismatch
16	VirNucPro	ML (MLP)	Refeq viral	Broad viral DNA detection (both prokaryotic + archaeal and eukaryotic viruses), prophage detection	1	- Requires valid CDS regions for accurate analysis - Potential false positives from bacterial MGEs (mobile genetic elements) - Six-frame translation may introduce non-authentic protein products - Limited by standard codon usage assumptions
17	DETIRE	ML (CNN + BiLSTM)	Refeq viral	DNA viruses (bacteriophages + archaeal and eukaryotic)	9	- Accuracy drops slightly on very long contigs (>3 kbp) - The embedding model is optimized for 500 bp fragments
18	VirRep	ML (BERT + BiLSTM)	Published human gut virome catalogs, genomes of human gut prokaryotes	DNA viruses (prokaryotic and eukaryotic)	4	- Comprehensive evaluation across diverse environments is lacking - Requires biome-specific fine-tuning for non-gut or other environments
19	ViraLM	ML (DNABERT-2 + binary classifier)	NCBI RefSeq viral, data from public studies	DNA and RNA viruses from bacterial, archaeal, and eukaryotic hosts	10	- May lose long-range genomic context - Provides limited functional information - Faces computational challenges when processing lengthy input sequences
20	GCNFrame	ML (MLP)	NCBI and ICEberg	Prokaryotic dsDNA phages and bacterial host elements (ICEs), Prophages	4	- Sensitive to hyperparameter settings, which can affect model stability - Graph memory overhead due to inclusion of all possible k-mers (even those with zero counts)
21	PhaGCN	ML (CNN + GCN)	NCBI RefSeq	Bacteriophages (Caudovirales); validated families: Myoviridae, Siphoviridae, Podoviridae, Ackermannviridae, Herelleviridae, Demerecviridae, extensible to Rudiviridae, Inoviridae	115	- Performance decreases with shorter contigs (<4 kb) - Caudovirales-centric (covers ≈ 95.8% of known phages, but is not universal)
22	geNomad	ML (DNN + XGBoost)	GTDB, TOPAZ, PLSDB, RefSeq, IMG/VR version 3, Specialized sets of viruses (Nucleocytoviricota, Leviviridae, Asgard archaea viruses, etc.)	Broad-spectrum DNA viruses (prokaryotic + eukaryotic) and plasmids, capable of identifying novel MGEs, sequences of plasmids, proviruses, RNA, and giant viruses	594	- May not classify completely novel viral families lacking known markers - At lower taxonomic ranks, it shows limited family coverage (61.8%)

**Table 4 viruses-17-01538-t004:** K-Mer-based tools for virus detection.

No.	Tool	Methodology	Database Source	Viral Specialization	CI	Limitations
**1**	Kraken2	k-mer	Refeq viral	Broad DNA and RNA viruses, dsDNA bacteriophages and vertebrate DNA viruses, divergent RNA viruses	6081	- Slightly reduced specificity due to probabilistic hashing - Limited species-level resolution for highly similar taxa - Relies on the quality and completeness of the reference database
**2**	DisCVR	k-mer	NCBI taxonomy database	Human DNA and RNA viruses	11	- Lower sensitivity for low-coverage or highly fragmented samples
**3**	CLARK	k-mer	NCBI/Refeq viral	Bacterial, archaeal, and eukaryotic viruses are mainly DNA viruses, but RNA viruses can be detected if represented in the database	775	- Precision–sensitivity tradeoff depending on k-mer size
**4**	Vista	Pairwise alignment + k-mer profiles + ML	NCBI Viral Genomes Resource	Optimized for Caudoviricetes—tailed dsDNA bacteriophages infecting bacterial and archaeal hosts; eukaryotic viral families, spanning ssDNA, dsDNA, ssRNA, and dsRNA	3	- Bias toward taxa with many reference sequences - Difficulty handling singletons and segmented genomes - Requires frequent database updates to maintain accuracy

## Data Availability

The data presented in this study is available on request from the corresponding author.

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
