# Peer review of "Bioinformatics Tools and Approaches for Virus Discovery in Genomic Data: A Systematic Review"

_viruses, 2025, doi:10.3390/v17121538_

Round 1

Reviewer 1 Report

Comments and Suggestions for Authors

In this review article, Galeeva and colleagues summarize and discuss a comprehensive body of literature about computational tools for virus identification in sequencing data and their taxonomic annotation. It provides an excellent overview of the principal approaches and available tools and will be a very valuable resource, in particular for students that enter the field.

To make it even more valuable, it would be great if the authors could also summarize performance metrics of the various tools. I realize that the authors did not “aim to provide benchmarking assessments” (line 673), but I would like to ask them to carefully reconsider this. From my experience, the choice and usage of a particular tool depends on several factors, which include the rate of false positives the user can expect (particularly but not exclusively important in the case of large DNA viruses with a considerable number of genes that have cellular homologs) and the expected rate of false negatives (particularly but not exclusively important in the case of highly divergent novel RNA virus lineages). I am sure that having such a comprehensive overview of the performance measures would be much appreciated by the community. I guess that the original publications of many of the tools discussed in this review should report these or similar performance metrics. A classification into performance categories (low, medium, high) would be also helpful. A similar reasoning applies to scalability/runtime assessments to give the reader an idea about the scale of data that can be analyzed with a particular tool. If the authors want to address this concern, these data on performance/scalability could be added to table A1 or form a new table.

The authors may want to consider adjusting the title of this review to reflect a strong focus on tools for virus identification/discovery from sequencing data and not “just” taxonomic annotation. Virus identification and taxonomic classification are of course connected, but there is another (large) set of bioinformatics tools (not discussed by the authors) that are exclusively dedicated to virus taxonomy.

The discussed tools and their underlying approaches are highly diverse and the input data may differ (nucleotide- or amino acid-based or both). This makes it difficult to understand the degree of sequence divergence (from the closest known reference virus) a particular tool/approach can deal with. Is there any data that could be added in this respect?

I am a bit surprised that GenSeed-HMM (PMID: 26973638) is not among the list of HMM-based tools (I am not one of the authors). Although GenSeed-HMM has a strong focus on viral genome assembly, it can be applied for virus identification and even classification if, for instance, virus family-specific HMMs are utilized.

Figure 3 is mislabeled as Figure 4 in the figure legend.

Consider to consistently use the term “amino acid sequence” instead of “protein sequence” throughout the review.

Author Response

Reviewer 1

In this review article, Galeeva and colleagues summarize and discuss a comprehensive body of literature about computational tools for virus identification in sequencing data and their taxonomic annotation. It provides an excellent overview of the principal approaches and available tools and will be a very valuable resource, in particular for students that enter the field.

1)To make it even more valuable, it would be great if the authors could also summarize performance metrics of the various tools. I realize that the authors did not “aim to provide benchmarking assessments” (line 673), but I would like to ask them to carefully reconsider this. From my experience, the choice and usage of a particular tool depends on several factors, which include the rate of false positives the user can expect (particularly but not exclusively important in the case of large DNA viruses with a considerable number of genes that have cellular homologs) and the expected rate of false negatives (particularly but not exclusively important in the case of highly divergent novel RNA virus lineages). I am sure that having such a comprehensive overview of the performance measures would be much appreciated by the community. I guess that the original publications of many of the tools discussed in this review should report these or similar performance metrics. A classification into performance categories (low, medium, high) would be also helpful. A similar reasoning applies to scalability/runtime assessments to give the reader an idea about the scale of data that can be analyzed with a particular tool. If the authors want to address this concern, these data on performance/scalability could be added to table A1 or form a new table.

Answer:  We sincerely appreciate the reviewer's thoughtful suggestion regarding the inclusion of performance metrics and benchmarking data for the various tools discussed in our review. We fully agree that such information would significantly enhance the practical value of this work for the research community, particularly for researchers needing to select appropriate tools.

We carefully considered this important aspect during the preparation of our manuscript. Unfortunately, conducting a comprehensive benchmarking study of all reviewed tools would require substantial computational, human resources, and time that are beyond the scope of our current project. Such an undertaking would involve:

  1. Establishing standardized benchmark datasets covering diverse viral families and viral types
  2. Running all tools under comparable conditions with consistent parameter settings
  3. Systematic evaluation across multiple performance dimensions (sensitivity, specificity, computational efficiency)

We recognize that some original publications report performance metrics, these are often evaluated using different datasets, metrics, and conditions, making direct comparisons potentially misleading without proper re-evaluation under standardized conditions.

As a compromise, we have:

  1. Where available from original publications, we noted key performance characteristics in our discussion of individual tools
  2. Add information about the advantages and limitations of each tool based on the original publication

We summarized key information (n = 19 parameters) for all 54 viral analysis tools based on their original publications in Supplementary Table 1. The summary included details such as methodology, database source, input type, advantages, limitations, etc. 

In the new version of our article, we have also added a summary section on the applicability of tools and created a guide for selecting viral analysis tools based on specific research goals and virus types on the original publication.

2) The authors may want to consider adjusting the title of this review to reflect a strong focus on tools for virus identification/discovery from sequencing data and not “just” taxonomic annotation. Virus identification and taxonomic classification are, of course connected, but there is another (large) set of bioinformatics tools (not discussed by the authors) that are exclusively dedicated to virus taxonomy.

 Answer: We have fixed this in the new version of the manuscript new title is ‘Bioinformatics tools and approaches for virus detection in genomic data: a systematic review’

3)The discussed tools and their underlying approaches are highly diverse and the input data may differ (nucleotide- or amino acid-based or both). This makes it difficult to understand the degree of sequence divergence (from the closest known reference virus) a particular tool/approach can deal with. Is there any data that could be added in this respect?

 Answer: We expanded the summary table and included this information (supplementary table1)  

4)I am a bit surprised that GenSeed-HMM (PMID: 26973638) is not among the list of HMM-based tools (I am not one of the authors). Although GenSeed-HMM has a strong focus on viral genome assembly, it can be applied for virus identification and even classification if, for instance, virus family-specific HMMs are utilized.

 Answer:  Thank you for your suggestion. GenSeed-HMM is indeed a valuable tool for targeted sequence assembly and can support viral genome reconstruction when virus-specific HMMs are applied. However, it was excluded from our review because its primary focus is progressive assembly, not automated viral identification or taxonomic classification, which were the main criteria for inclusion.

5)Figure 3 is mislabeled as Figure 4 in the figure legend.

   Answer: Thank you for your suggestion. We have fixed this issue 

6) Consider to consistently use the term “amino acid sequence” instead of “protein sequence” throughout the review.

    Answer: Thank you for your suggestion. We have fixed this issue 

Reviewer 2 Report

Comments and Suggestions for Authors

General Comments

The manuscript addresses an important and timely subject: the systematic review of bioinformatics tools for viral taxonomic annotation. With the rapid increase in metagenomic data, the review is highly relevant and useful for researchers selecting appropriate methods.

The structure (Abstract, Introduction, Methods, Results, Discussion, Tables) is logical and follows PRISMA guidelines. However, certain sections are too dense with technical details (e.g., machine learning methods) and could benefit from more synthesis rather than listing.

The manuscript provides a broad overview, but the critical evaluation of tools is somewhat limited. A more explicit comparison of strengths, weaknesses, and applicability of each category of tools would add more value than descriptive summaries.

The discussion rightly highlights the lack of benchmarking and reproducibility issues. However, the implications for practical tool selection (e.g., which approach is best under different dataset constraints) are underdeveloped.

The limitations section is good, but it should explicitly discuss bias in tool coverage (PubMed-only search, English-language bias, exclusion of unpublished but widely used tools). Aditionally, the literature search was restricted to PubMed. While PubMed is comprehensive for biomedical and life sciences, it does not cover all relevant interdisciplinary journals. I suggest that the authors consider using Web of Science as their primary database (or at least in combination with PubMed), since WoS provides broader, multidisciplinary coverage and citation tracking, and would likely capture additional computational and bioinformatics tools not indexed in PubMed.

In the Materials and Methods, the authors state that no restrictions were imposed on the year of publication. However, viral metagenomic data has only become widely available since the introduction of next-generation sequencing (e.g., Illumina in 2008, Ion Torrent in 2012). Could the authors clarify the rationale for including earlier publications, given that relevant computational tools are unlikely to have been developed before this period?

Specific comments:

  • Lines 149–158
    Timeline figure is clear, but the text mostly repeats what’s already in the figure. Suggest reducing redundancy and emphasizing insights (e.g., “shift from alignment-based to AI methods after 2015”). Also, include what technology was useful to produce metagenomic data available (Illumina, Ion Torrent, PacBio, etc…)  
  •  Lines 649–662
    The point about hybrid approaches is strong—could be expanded into a subsection summarizing future trends.
  • Lines 718–726
    Important observation on open code. Suggest emphasizing reproducibility crisis in computational virology.

Author Response

Reviewer 2

General Comments

The manuscript addresses an important and timely subject: the systematic review of bioinformatics tools for viral taxonomic annotation. With the rapid increase in metagenomic data, the review is highly relevant and useful for researchers selecting appropriate methods.

  1. The structure (Abstract, Introduction, Methods, Results, Discussion, Tables) is logical and follows PRISMA guidelines. However, certain sections are too dense with technical details (e.g., machine learning methods) and could benefit from more synthesis rather than listing.

Answer: 

We appreciate the comment. The machine learning section covers many different methods, so we described it in more detail to show the variety of approaches used in these tools.

  1. The manuscript provides a broad overview, but the critical evaluation of tools is somewhat limited. A more explicit comparison of strengths, weaknesses, and applicability of each category of tools would add more value than descriptive summaries.

Answer: We summarized key information (n = 19 parameters) for all 54 viral analysis tools based on their original publications in Supplementary Table 1. The summary included details such as methodology, database source, input type, advantages, limitations and etc. 

  1. The discussion rightly highlights the lack of benchmarking and reproducibility issues. However, the implications for practical tool selection (e.g., which approach is best under different dataset constraints) are underdeveloped.

Answer: Using this information from the original publications , we created a roadmap to guide tool selection according to research goals and virus types presented in Figure 4.

  1. The limitations section is good, but it should explicitly discuss bias in tool coverage (PubMed-only search, English-language bias, exclusion of unpublished but widely used tools). Aditionally, the literature search was restricted to PubMed. While PubMed is comprehensive for biomedical and life sciences, it does not cover all relevant interdisciplinary journals. I suggest that the authors consider using Web of Science as their primary database (or at least in combination with PubMed), since WoS provides broader, multidisciplinary coverage and citation tracking, and would likely capture additional computational and bioinformatics tools not indexed in PubMed.

Answer: We appreciate this comment and acknowledge that using a single database may introduce some bias. We chose PubMed because it provides the most comprehensive and standardized coverage of peer-reviewed literature in virology, microbiology, and bioinformatics, which are the primary focus areas of this review. PubMed structured indexing, consistent MeSH terminology, and direct linkage to openly available full-text articles made it particularly suitable for ensuring methodological transparency and reproducibility. While we recognize that databases such as Web of Science offer broader multidisciplinary coverage, our objective was to capture tools with demonstrated relevance and validation within the biomedical and virological research community, for which PubMed remains the most reliable and widely used source.

5)In the Materials and Methods, the authors state that no restrictions were imposed on the year of publication. However, viral metagenomic data has only become widely available since the introduction of next-generation sequencing (e.g., Illumina in 2008, Ion Torrent in 2012). Could the authors clarify the rationale for including earlier publications, given that relevant computational tools are unlikely to have been developed before this period?

Answer: Thank you for your comment. It is correct that viral metagenomic data became widely available only after the introduction of next-generation sequencing technologies (Illumina in 2008, Ion Torrent in 2012). However, we deliberately chose not to restrict publication years for several important reasons. Tools like BLAST (1990), ClustalW (1994), and PSI-BLAST (1997) represent foundational sequence comparison algorithms that, while developed before the NGS era, have been continuously adapted and remain integral components of modern viral annotation pipelines. Understanding their development provides crucial context for how current tools evolved. Many fundamental algorithms and approaches used in modern viral taxonomic annotation tools have their roots in pre-NGS bioinformatics. For instance, Hidden Markov Models, multiple sequence alignment algorithms, and k-mer-based approaches were established before 2008 but have been successfully adapted for NGS data analysis. Several tools developed in the early 2000s (like MUSCLE 2004, Phage Finder 2006) bridged the transition from Sanger sequencing to NGS and were subsequently updated to handle larger-scale metagenomic datasets. These tools demonstrate the adaptability of core methodologies to new data types. Our systematic review aims to document the complete trajectory of computational approaches in viral taxonomy, showing how methods evolved from analyzing individual sequences to handling millions of metagenomic reads. This historical perspective helps readers understand current tool capabilities and limitations.

Specific comments:

  • Lines 149–158
    6) Timeline figure is clear, but the text mostly repeats what’s already in the figure. Suggest reducing redundancy and emphasizing insights (e.g., “shift from alignment-based to AI methods after 2015”). Also, include what technology was useful to produce metagenomic data available (Illumina, Ion Torrent, PacBio, etc…)  
  • Answer: Thank you for your suggestion. We added this information to the figure 2.
  • 7)  Lines 649–662
    The point about hybrid approaches is strong—could be expanded into a subsection summarizing future trends.
  • Answer: Thank you for your suggestion. We decided not to change the structure of the review, but paid special attention to combined approaches in the discussion section.
  • Lines 718–726
    8)Important observation on open code. Suggest emphasizing reproducibility crisis in computational virology.
  • Answer:  Thank you for your suggestion. we added this information to discussion section lines  786-790

Round 2

Reviewer 1 Report

Comments and Suggestions for Authors

I do not have further comments.

Author Response

We thank the reviewer for their valuable feedback. We have improved consistency of the narrative and text flow by replacing "viral taxonomic/taxonomy" terminology with "discovery/identification" throughout the manuscript and enhanced the figures for better clarity

Reviewer 2 Report

Comments and Suggestions for Authors

The authors have responded to all queries and details raised during the peer review process. 

Author Response

We thank the reviewer for their valuable feedback. We have improved consistency of the narrative and text flow by replacing "viral taxonomic/taxonomy" terminology with "discovery/identification" throughout the manuscript and enhanced the figures for better clarity.